# Comparison of the Effects of Sugammadex, Neostigmine, and Pyridostigmine on Postoperative Nausea and Vomiting: A Propensity Matched Study of Five Hospitals

**DOI:** 10.3390/jcm9113477

**Published:** 2020-10-28

**Authors:** Jong Ho Kim, Man-Sup Lim, Jun Woo Choi, Haewon Kim, Young-Suk Kwon, Jae Jun Lee

**Affiliations:** 1Department of Anesthesiology and Pain Medicine, Chuncheon Sacred Heart Hospital, College of Medicine, Hallym University, Chuncheon 24253, Korea; ellemes@hallym.ac.kr (J.H.K.); jjun622@hallym.or.kr (J.W.C.); haewon@hallym.or.kr (H.K.); 2Institute of New Frontier Research Team, Hallym University, Chuncheon 24253, Korea; 3Department of medical education, college of medicine, Hallym University, Chuncheon 24253, Korea; poik99@hallym.or.kr

**Keywords:** sugammadex, neostigmine, pyridostigmine, postoperative nausea and vomiting

## Abstract

Thus far, few studies have compared the effects of sugammadex and cholinesterase inhibitors on postoperative nausea and vomiting (PONV), and the results have been controversial. Here, we compared the effects of sugammadex, neostigmine, and pyridostigmine on PONV by means of a five hospital analysis with propensity score matching. We analyzed adults aged ≥ 18 years who underwent general anesthesia between January 2014 and December 2019. Following propensity score matching, 7793 patients were included in each of the neostigmine and sugammadex matched patient groups (absolute standardized difference (ASD), 0.01–0.07), and 10,197 patients were included in each of the pyridostigmine and sugammadex matched patient groups (ASD, 0.01–0.02), while 19,377 patients were included in each of the pyridostigmine and neostigmine matched patient groups. (ASD, 0.01–0.19). The odds of PONV were low in the sugammadex group (odds ratio, 0.65; 95% confidence interval, 0.59–0.72; *p* < 0.0001) and pyridostigmine group (odds ratio, 0.22; 95% confidence interval, 0.20–0.24; *p* < 0.0001) compared to the neostigmine group, while there was no difference between sugammadex and pyridostigmine (odds ratio, 0.95; 95% confidence interval, 0.86–1.04; *p* = 0.281). Therefore, sugammadex and pyridostigmine may lower the incidence of PONV compared to neostigmine in patients undergoing general anesthesia.

## 1. Introduction

Postoperative nausea and vomiting (PONV) is not usually a fatal postoperative complication but is nonetheless particularly distressing [1]. There have been many studies on the risk factors for PONV development and remission [2,3,4,5,6,7]. Notably, high-dose neuromuscular block reversal has been reported to increase PONV [8], but the effect of neuromuscular block reversal on PONV remains controversial [9].

Neuromuscular relaxation is an essential component of general anesthesia. In most instances, neuromuscular blockade is reversed using acetylcholine esterase inhibitors [10]. Cholinesterase inhibitors stimulate muscarinic receptors to increase motility and secretions in the esophagus, stomach, and small and large intestine. However, cholinesterase inhibitors can cause vomiting and diarrhea. Muscarinic antagonists are commonly administered to mitigate the effects of excess acetylcholine during reversal of neuromuscular relaxation [11].

Sugammadex, a modified γ-cyclodextrin, selectively binds to steroidal neuromuscular blockers (such as rocuronium and vecuronium) to encapsulate and inactivate neuromuscular blockers during reversal of neuromuscular blockade. The resulting inactive complex is eliminated from the body, in accordance with the pharmacokinetic properties of sugammadex [12,13]. In contrast to acetylcholine esterase inhibitors, the muscarinic effect of sugammadex is limited. In this propensity score matching study, we compared sugammadex with commonly used acetylcholine esterase inhibitors, pyridostigmine, and neostigmine, in terms of the effect on PONV. Additionally, neostigmine and pyridostigmine were compared.

## 2. Patients and Methods

### 2.1. Patients

This study was approved by the Clinical Research Ethics Committee of Chuncheon Sacred Heart Hospital, Hallym University (IRB No. 2020-07-010). All data were acquired from the clinical data warehouse of Hallym University Medical Center, which is a database of medical records (including diagnoses), prescriptions, and test results. All patients in this study were at least 18 years of age and had undergone surgery under general anesthesia at one of the five hospitals of Hallym University between January 2014 and December 2019. The exclusion criteria were as follows: Repeat surgery within 24 h after anesthesiaLack of consciousness after surgeryVentilator therapy after anesthesiaNausea or vomiting before anesthesiaInsufficient data because the follow-up duration was shorter than 24 hNeuromuscular block reversal drugs (NMBRD; sugammadex (Bridion, Kenilworth, NJ, USA), neostigmine (Neostigmine Methylsulfate Injection Daihan, Seoul, Korea), and pyridostigmine (Pyridomine, Seoul, Korea) are not used.Two or more types of NMBRD are used.

### 2.2. PONV, Neuromuscular Block Reversal Drugs and Covariates

PONV was defined as nausea or vomiting within 24 h after surgery and anesthesia. The NMBRD used during recovery from general anesthesia was neostigmine, pyridostigmine, or sugammadex. Covariates included age, female sex, obesity (body mass index ≥ 30 kg/m^2^), emergency status, anesthesia duration, American Society of Anesthesiologists physical status, use of N_2_O, use of inhalation anesthetics during anesthesia maintenance, patient-controlled analgesia after surgery, intraoperative transfusion, history of diabetes and smoking, intraoperative use of antiemetics and opioids, and type of surgery (intraabdominal, gynecologic, or otorhinolaryngology surgery).

### 2.3. Statistics

Continuous data are presented as median and interquartile range because the data did not exhibit a normal distribution. Categorical data are presented as frequencies and percentages. The odds ratios (unadjusted, adjusted for all variables, adjusted for all variables and propensity scores) and 95% confidence intervals for the occurrence of PONV within 24 h after surgery were determined by means of logistic regression analysis.

Propensity scores were calculated based on the NMBRD type. Selection bias was reduced by using the absolute standardized difference (ASD). Groups are generally considered similar when the standardized difference is less than 20% [14]. The rationale and methods for using propensity scores when analyzing the exposure variable have been described elsewhere [15,16]. We performed two propensity score matching analyses: sugammadex vs. neostigmine, sugammadex vs. pyridostigmine and neostigmine vs. pyridostigmine. All *p*-values were two-sided, and a *p*-value < 0.05 was considered indicative of statistical significance. IBM SPSS Statistics (version 26.0; IBM Corp., Armonk, NY, USA) was used for the statistical analyses.

## 3. Results

### 3.1. Patient Characteristics

Figure 1 shows the flow chart of the study progress. From January 2014 to December 2019, 168,641 patients underwent general anesthesia at one of the five hospitals of Hallym University. In total, 25,270 patients met at least one exclusion criterion. Thus, 143,371 patients were initially included in the study. Subsequently, 282 patients were excluded due to missing data, while 17,993 patients were excluded because they used no NMBRDs or more than two types of NMBRDs. Finally, 125,096 patients were included in the analysis. Of these patients, 10,197 received sugammadex, 33,421 received neostigmine, and 81,478 received pyridostigmine. Table 1 (neostigmine and sugammadex), Table 2 (pyridostigmine and sugammadex), and Table 3 (neostigmine and pyridostigmine) show the baseline patient characteristics and clinical data, before and after matching. Before matching of the neostigmine and sugammadex groups, the ASD ranged from 0.00 to 0.49 across the variables. After matching, the ASD was ≤0.07 for all variables. Before matching of the pyridostigmine and sugammadex groups, the ASD ranged from 0.04 to 0.73 across the variables. After matching, the ASD was ≤0.02 for all variables. Before matching of the neostigmine and pyridostigmine groups, the ASD ranged from 0.04 to 3.61 across the variables. After matching, the ASD was ≤0.19 for all variables.

### 3.2. Occurrence and Odds Ratio of PONV

The number of PONV cases before propensity score matching was 8228 (10.1%) in the pyridostigmine group, 4626 (13.8%) in the neostigmine group, and 972 (9.5%) in the sugammadex group. After matching between the neostigmine and sugammadex groups, the number of PONV cases was 1108 (14.2%) in the neostigmine group and 766 (9.8%) in the sugammadex group. After matching between the pyridostigmine and sugammadex groups, the number of PONV was 994 (9.7%) in the pyridostigmine group and 972 (9.5%) in the sugammadex group. After matching between the neostigmine and pyridostigmine groups, the number of PONV was 994 (14.4%) in the neostigmine group and 972 (3.7%) in the pyridostigmine group.

The unadjusted and adjusted odds ratios (OR) for PONV of the sugammadex group between the sugammadex and neostigmine groups were <1.0 and were both statistically significant (unadjusted, OR: 0.66, 95% confidence interval [CI]: 0.60–0.73, *p* < 0.0001; adjusted for all variables, OR: 0.65, 95% CI: 0.59–0.72, *p* < 0.0001; adjusted for all variables and propensity scores, OR: 0.65, 95% CI: 0.59–0.72, *p* < 0.0001). The unadjusted and adjusted OR for the occurrence of PONV of the sugammadex group between the sugammadex and pyridostigmine groups were <1.0 but were not statistically significant (unadjusted, OR: 0.98, 95% CI: 0.89–1.07, *p* = 0.602; adjusted for all variables, OR: 0.96, 95% CI: 0.87–1.05, *p* = 0.353; adjusted for all variables and propensity scores, OR: 0.95, 95% CI: 0.86–1.04, *p* = 0.281). The unadjusted and adjusted odds ratios OR for PONV of the pyridostigmine group between the neostigmine and pyridostigmine groups were <1.0 and were both statistically significant (unadjusted, OR: 0.23, 95% CI: 0.21–0.25, *p* < 0.0001; adjusted for all variables, OR: 0.22, 95% CI: 0.20–0.24, *p* < 0.0001, adjusted for all variables and propensity scores, OR: 0.22, 95% CI: 0.20–0.24, *p* < 0.0001).

## 4. Discussion

This large-scale, five hospital study compared the likelihood of PONV between sugammadex and cholinesterase inhibitors through propensity score matching. The odds of PONV were lower in the sugammadex group and pyridostigmine group than in the neostigmine group, but did not significantly differ between the sugammadex and pyridostigmine groups. 

Several studies have compared sugammadex with conventional cholinesterase inhibitors in terms of the likelihood of PONV [12,17,18,19,20]. Yağan et al. reported that sugammadex was associated with a lower incidence of PONV within the first 1 h postoperatively compared to neostigmine, and less antiemetic agent use within the first 24 h postoperatively [12]. In a systematic review, Hristovska et al. found that sugammadex was associated with a lower risk of PONV compared to neostigmine, and Gan Tong et al. suggested the use of sugammadex instead of neostigmine as NMBRD as a strategy to reduce the baseline risk of PONV [20,21]. In a meta-analysis of Hristovska et al., including 10 randomized controlled trials, was performed to compare the efficacy and safety of sugammmadex and neostigmine regardless of dose, and 6 of 10 randomized controlled trials were included for comparative analysis of PONV. In their study, patients receiving sugammadex had 40% fewer adverse effects than those receiving neostigmine. In PONV, the relative risk of sugammedex to neostigmine was 0.52 (95% CI, 0.28–0.97). The results were similar to those of our study. However, due to the small sample size and limited evidence for the six cases included in the meta-analysis, the authors of the study suggested that a study with a large sample size and low risk of bias is needed in the future. Our study confirms and supports their findings by large sample size and reducing the bias with propensity score matching analysis.

Tas Tuna et al. reported no significant difference in the risk of PONV between sugammadex and neostigmine [17]. Peach et al. also reported that sugammadex did not reduce PONV compared to neostigmine [18]. However, because the previous studies were conducted in non-elderly patients who were in good general condition and had undergone laparoscopic resection, it may be difficult to generalize their findings to other populations. Lee et al. reported that the use of sugammadex was beneficial in terms of postoperative vomiting and the requirements for antiemetics compared to pyridostigmine [19]. Their study also used propensity score matching, but their findings were substantially different from the results of our study. Notably, they targeted patients who received patient-controlled analgesia after surgery, which is a known risk factor for PONV [6,22]. This might have contributed to the discrepancy in the results. Kang et al. [23] reported that sugammadex did not reduce the incidence of PONV compared to pyridostigmine in patients undergoing microvascular decompression, and the incidence of PONV decreased when administered simultaneously with panlonosetrone. It would be difficult to consider as a standalone effect of sugammadex.

There have been many clinical studies and reviews concerning the effects of neostigmine on PONV. However, the results are heterogenous among studies. In one meta-analysis, the use of neostigmine did not increase the risk of PONV, but those findings might not have been clinically meaningful because of the limited number of patients included [9]. The results of randomized controlled trials have also implied that neostigmine does not increase the incidence or severity of PONV [24,25]. However, other studies have demonstrated that high doses of neostigmine can increase the risk of PONV [8,25,26]. Nonetheless, the neostigmine dose has not been found to significantly increase the incidence of PONV [9]. Pyridostigmine can cause PONV due to muscarinic side effects [23]. A relative overdose of pyridostigmine compared to anticholinergic drugs can also trigger PONV [19].

In this study, PONV occurrence of neostigmine was higher than the other two drugs. The difference of PONV occurrence between neostigmine and the other two drugs is difficult to consider as a difference in direct drug action mechanisms, because the neostigmine and pyridostigmine are similar in structure, and pyridostigmine shares neostigmine covalent binding to acetylcholinesterase and its lipid solubility [10,11]. However, some factors may cause the difference between neostigmine and the other two drugs for PONV. First, anticholinergic drugs administered concurrently with NMBRD may affect differences of PONV occurrence [9]. In this study, because sugammadex was not administered simultaneously with an anticholinergic agent, it was not investigated separately, but the preference of an anticholinergic agent administered at the same time may affect the difference in PONV. Atropine has been reported to have an antiemetic effect [25,27]. Second, because high doses of neostigmine may induce PONV [8,25,26], the administered dose of the drugs may affect the difference in the occurrence of PONV. Some clinicians frequently use full doses due to fast onset and short action time of neostigmine [10]. However, more research is needed on the factors that may cause differences in the occurrence of PONV between the two drugs. 

The effects of sugammadex and cholinesterase inhibitors on PONV can also be considered in terms of subsequent serious postoperative complications and death. Vomiting after surgery places patients at risk of aspiration, which is closely associated with pneumonia. Notably, elderly patients and those with poor performance status have a high risk of aspiration [28,29]. The mortality rate for aspiration pneumonia is largely dependent on the volume and content of aspirate, and can reach 70% [30]. Compared to cholinesterase inhibitors, sugammadex can more quickly reverse neuromuscular blockade, regardless of the depth of the block. Sugammadex also reduces the risk of residual paralysis after surgery [20]. Recovery of muscle tone affects the diaphragm, upper respiratory strength, and chest wall strength, and may also improve coughing ability, attenuate secretions, reduce alveolar collapse (associated with pneumonia), and prevent microaspiration [31,32]. A recent multicenter matched cohort study showed that sugammadex can reduce the likelihood of postoperative pneumonia compared to neostigmine [33].

There were some limitations to this study. First, vomiting and nausea were not analyzed separately. However, although vomiting and nausea have different mechanisms of development [34,35], there is no clinically meaningful difference in treatment for patients with vomiting and those with nausea. Moreover, nausea is regarded as a precursor to vomiting [36]. Second, the effects of the dose of NMBRD were not analyzed. There is no established method for comparing the effects of sugammadex and other cholinesterase inhibitors based on dose. Furthermore, the dose used varies depending on the type of neuromuscular blockade and patient body weight. Additional studies are needed concerning the relationship between the dose of sugammadex and likelihood of PONV. Third, it may be important to categorize PONV as early or delayed onset because risk factors may differ between these two types of PONV [9]. Considering the duration of action of NMBRDs, they are likely to have a greater impact on early PONV [12]. However, the exact time of PONV occurrence could not be determined from our data.

## 5. Conclusions

In this study, multivariate analysis based on propensity score matching showed that the use of sugammadex and pyridostigmine was associated with a lower incidence of PONV compared to neostigmine. In contrast, the incidence of PONV did not significantly differ between patients receiving sugammadex and those receiving pyridostigmine. This study supports and expands on the findings of previous studies reporting that sugammadex reduced the incidence of PONV compared to neostigmine. However, our findings differ from previous studies with respect to pyridostigmine. Our findings should be confirmed in prospective large-scale randomized controlled studies.

## Figures and Tables

**Figure 1 jcm-09-03477-f001:**
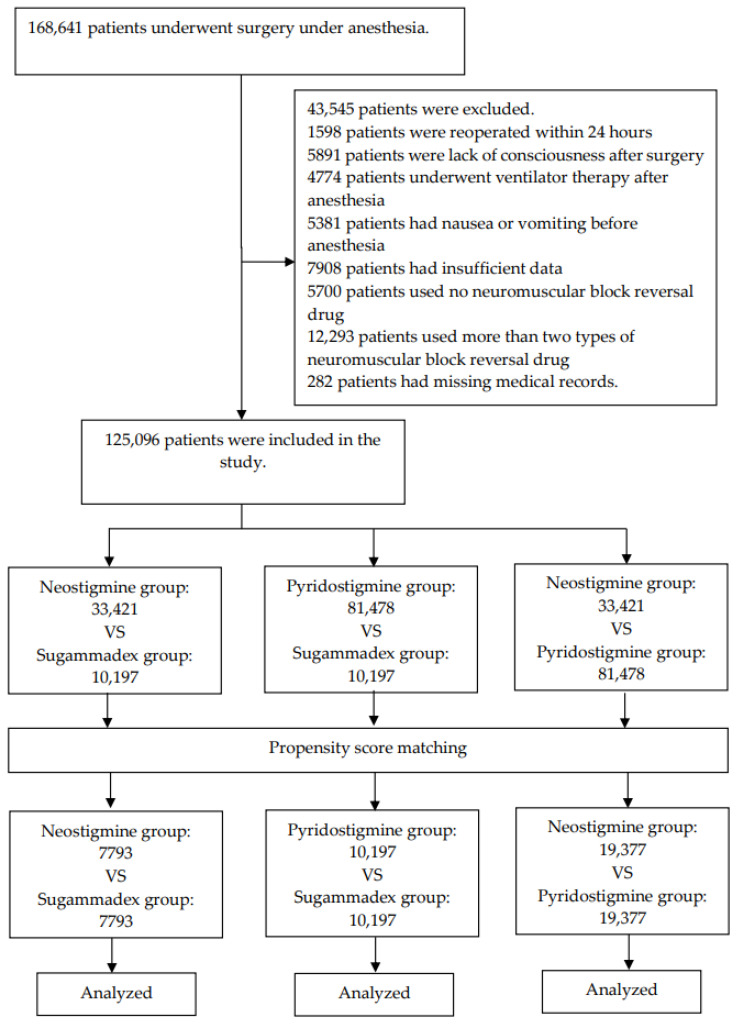
Flow chart.

**Table 1 jcm-09-03477-t001:** Baseline patient characteristics and clinical data before and after matching: comparison between the neostigmine and sugammadex groups.

		Before Matching			After Matching		
		Neostigmine (*n* = 33,421)	Sugammadex (*n* = 10,197)	ASD	Neostigmine (*n* = 7793)	Sugammadex (*n* = 7793)	ASD
**Demographic data**							
Age (years, median (IQR))		64 (39, 52)	61 (48, 72)	0.45	60 (49, 72)	62(50, 73)	0.05
Female sex (*n*, %)		18,079 (54.1)	4338 (42.5)	0.23	3635 (46.6)	3471 (44.5)	0.07
Obesity (BMI ≥ 30 kg/m^2^) (*n*, %)		2397 (7.2)	924 (9.1)	0.07	692 (8.9)	644 (8.3)	0.04
**Type of surgery**							
Intraabdominal surgery (*n*, %)		8820 (26.4)	1943 (19.1)	0.19	1647 (21.1)	1658 (21.3)	0.00
GY surgery (*n*, %)		2758 (8.3)	239 (2.3)	0.39	299 (3.8)	236 (3)	0.05
ENT surgery (*n*, %)		1351 (4)	1346 (13.2)	0.27	771 (9.9)	841 (10.8)	0.03
**Anesthesia-related factors**							
ASA PS	1	8534 (25.5)	1943 (19.1)	0.14	1428 (18.3)	1156 (15.0)	0.02
(*n*, %)	2	16,415 (49.1)	5517 (54.1)		3539 (45.4)	4305 (55.2)	
	3	7978 (23.9)	2362 (23.2)		2687 (34.5)	1969 (25.3)	
	4	491 (1.5)	373 (3.7)		139 (1.8)	352 (4.5)	
	5	3 (0)	2 (0)		0 (0)	1 (0)	
Anesthesia time (hours, median (IQR))		1.9 (1.3, 2.9)	1.9 (1.1, 3.5)	0.13	2.3 (1.5, 3.6)	2.2 (1.2, 3.9)	0.01
Inhalation anesthetics (*n*, %)		32,575 (97.5)	9461 (92.8)	0.18	7312 (93.8)	7288 (93.5)	0.01
N_2_O (*n*, %)		14 (0)	2009 (19.7)	0.49	14 (0.2)	16 (0.2)	0.00
Opioid (*n*, %)		33,325 (99.7)	9805 (96.2)	0.19	7718 (99)	7682 (98.6)	0.02
**Other factors**							
DM (*n*, %)		4289 (12.8)	1992 (19.5)	0.17	1559 (20)	1600 (20.5)	0.01
Current smoker (*n*, %)		5250 (15.7)	1821 (17.9)	0.06	1265 (16.2)	1329 (17.1)	0.02
Emergency (*n*, %)		6508 (19.5)	1426 (14)	0.16	1109 (14.2)	1087 (13.9)	0.02
Steroid (*n*, %)		1150 (3.4)	916 (9)	0.19	473 (6.1)	481 (6.2)	0.00
Transfusion (*n*, %)		1098 (3.3)	334 (3.3)	0.00	290 (3.7)	277 (3.6)	0.01
Antiemetics (*n*, %)		15,426 (46.2)	6987 (68.5)	0.48	4725 (60.6)	4729 (60.7)	0.00
PCA (*n*, %)		14,682 (43.9)	5568 (54.6)	0.21	4261 (54.7)	4228 (54.3)	0.01

IQR, interquartile range; BMI, body mass index; ASA PS, American Society of Anesthesiologists physical status; PCA, patient-controlled analgesia; DM, diabetes; GY, gynecologic; ENT, ear-nose-throat, ASD, absolute standardized difference.

**Table 2 jcm-09-03477-t002:** Baseline patient characteristics and clinical data before and after matching: comparison between the pyridostigmine and sugammadex groups.

		Before Matching			After Matching		
		Pyridostigmine (*n* = 81,478)	Sugammadex (*n* = 10,197)	ASD	Pyridostigmine (*n* = 10,197)	Sugammadex (*n* = 10,197)	ASD
**Demographic data**							
Age (years, median (IQR))		49 (36, 61)	61 (48, 72)	0.61	61 (49, 72)	61 (48, 72)	0.00
Female sex (*n*, %)		41,752 (51.2)	4338 (42.5)	0.18	4340 (42.6)	4338 (42.5)	0.01
Obesity (BMI ≥ 30 kg/m^2^) (*n*, %)		6353 (7.8)	924 (9.1)	0.04	887 (8.7)	924 (9.1)	0.00
**Type of surgery**							
Intraabdominal surgery (*n*, %)		13,670 (16.8)	1943 (19.1)	0.06	1862 (18.3)	1943 (19.1)	0.02
GY surgery (*n*, %)		10,894 (13.4)	239 (2.3)	0.73	228 (2.2)	239 (2.3)	0.01
ENT surgery (*n*, %)		12,077 (14.8)	1346 (13.2)	0.05	1373 (13.5)	1346 (13.2)	0.01
**Anesthesia-related factors**							
ASA PS	1	36,062 (44.3)	1943 (19.1)	0.62	1911 (18.7)	1943 (19.1)	0.01
(*n*, %)	2	37,908 (46.5)	5517 (54.1)		5513 (54.1)	5517 (54.1)	
	3	7142 (8.8)	2362 (23.2)		2564 (25.1)	2362 (23.2)	
	4	366 (0.4)	373 (3.7)		209 (2)	373 (3.7)	
	5	0 (0)	2 (0)		0 (0)	2 (0)	
Anesthesia time (hours, median (IQR))		1.8 (1.2, 2.6)	1.9 (1.1, 3.5)	0.27	2.1 (1.3, 3.3)	1.9 (1.1, 3.5)	0.01
Inhalation anesthetics (*n*, %)		77,576 (95.2)	9461 (92.8)	0.09	9413 (92.3)	9461 (92.8)	0.02
N_2_O (*n*, %)		18,431 (22.6)	2009 (19.7)	0.07	2100 (20.6)	2009 (19.7)	0.02
Opioid (*n*, %)		73,008 (89.6)	9805 (96.2)	0.34	9832 (96.4)	9805 (96.2)	0.01
**Other factors**							
DM (*n*, %)		4289 (12.8)	1992 (19.5)	0.17	1559 (20)	1600 (20.5)	0.01
Current smoker (*n*, %)		5250 (15.7)	1821 (17.9)	0.06	1265 (16.2)	1329 (17.1)	0.02
Emergency (*n*, %)		6508 (19.5)	1426 (14)	0.16	1109 (14.2)	1087 (13.9)	0.02
Steroid (*n*, %)		9818 (12)	1992 (19.5)	0.19	1945 (19.1)	1992 (19.5)	0.01
Transfusion (*n*, %)		15,891 (19.5)	1821 (17.9)	0.04	1748 (17.1)	1821 (17.9)	0.02
Antiemetics (*n*, %)		47,313 (58.1)	6987 (68.5)	0.23	7048 (69.1)	6987 (68.5)	0.01
PCA (*n*, %)		37,584 (46.1)	5568 (54.6)	0.17	5510 (54)	5568 (54.6)	0.01

IQR, interquartile range; BMI, body mass index; ASA PS, American Society of Anesthesiologists physical status; PCA, patient-controlled analgesia; DM, diabetes; GY, gynecologic; ENT, ear-nose-throat, ASD, absolute standardized difference.

**Table 3 jcm-09-03477-t003:** Baseline patient characteristics and clinical data before and after matching: comparison between the neostigmine and pyridostigmine.

		Before Matching			After Matching		
		Neostigmine (*n* = 33,421)	Pyridostigmine (*n* = 81,478)	ASD	Neostigmine (*n* = 19,377)	Pyridostigmine (*n* = 19,377)	ASD
**Demographic data**							
Age (years, median (IQR))		64 (39, 52)	49 (36, 61)	0.16	51 (38, 62)	51 (38, 63)	0.03
Female sex (*n*, %)		18,079 (54.1)	41,752 (51.2)	0.06	10,205 (52.7)	10,590 (54.7)	0.04
Obesity (BMI ≥ 30 kg/m^2^) (*n*, %)		2397 (7.2)	6353 (7.8)	0.05	1403 (7.2)	1544 (8.0)	0.06
**Type of surgery**							
Intraabdominal surgery (*n*, %)		8820 (26.4)	13,670 (16.8)	0.32	4256 (22.0)	4158 (21.5)	0.02
GY surgery (*n*, %)		2758 (8.3)	10,894 (13.4)	0.30	2086 (10.8)	2358 (12.2)	0.08
ENT surgery (*n*, %)		1351 (4)	12,077 (14.8)	0.78	1184 (6.1)	1148 (5.9)	0.02
**Anesthesia-related factors**							
ASA PS	1	8534 (25.5)	36,062 (44.3)	0.53	6188 (3193)	6073 (31.3)	0.01
(*n*, %)	2	16,415 (49.1)	37,908 (46.5)		9772 (50.4)	10,174 (52.5)	
	3	7978 (23.9)	7142 (8.8)		3245 (16.7)	2959 (15.3)	
	4	491 (1.5)	366 (0.4)		172 (0.9)	171 (0.9)	
	5	3 (0)	0 (0)		0 (0)	0 (0)	
Anesthesia time (hours, median (IQR))		1.9 (1.3, 2.9)	1.8 (1.2, 2.6)	0.20	1.9 (1.3, 2.9)	1.9 (1.3, 2.9)	0.04
Inhalation anesthetics (*n*, %)		32,575 (97.5)	77,576 (95.2)	0.36	18,840 (97.2)	18,994 (98.0)	0.19
N_2_O (*n*, %)		14 (0)	18,431 (22.6)	3.61	14 (0.1)	12 (0.1)	0.09
Opioid (*n*, %)		33,325 (99.7)	73,008 (89.6)	2.04	19,282 (99.5)	19,294 (99.6)	0.07
**Other factors**							
DM (*n*, %)		4289 (12.8)	9818 (12)	0.04	2349 (12.1)	2496 (12.9)	0.04
Current smoker (*n*, %)		5250 (15.7)	15,891 (19.5)	0.14	3424 (16.2)	3477 (17.9)	0.01
Emergency (*n*, %)		6508 (19.5)	14,712 (18.1)	0.05	3739 (19.3)	3636 (18.2)	0.02
Steroid (*n*, %)		9818 (12)	1992 (19.5)	0.19	1945 (19.1)	1992 (19.5)	0.01
Transfusion (*n*, %)		1098 (3.3)	1260 (1.5)	0.42	420 (3.7)	482 (3.6)	0.08
Antiemetics (*n*, %)		15,426 (46.2)	47,313 (58.1)	0.26	9523 (49.1)	8881 (45.8)	0.07
PCA (*n*, %)		14,682 (43.9)	37,584 (46.1)	0.05	4261 (46.2)	4228 (42.9)	0.07

IQR, interquartile range; BMI, body mass index; ASA PS, American Society of Anesthesiologists physical status; PCA, patient-controlled analgesia; DM, diabetes; GY, gynecologic; ENT, ear-nose-throat, ASD, absolute standardized difference.

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
