# Peer review of "Comparison of the Effects of Sugammadex, Neostigmine, and Pyridostigmine on Postoperative Nausea and Vomiting: A Propensity Matched Study of Five Hospitals"

_jcm, 2020, doi:10.3390/jcm9113477_

Round 1

Reviewer 1 Report

Dear Authors, 

I am pleased to review your article and I admire your effort considering the enormous amount of data you processed. I would suggest to improve tables layout, because are really difficult to read and to interpret.

In my opinion, to better read and understand tables, you should sort your data by demographic characteristics, type of surgery and potential risk factors for PONV following Apfel score. In this way, a reader may understand quickly the differences between sugammadex, pyridostigmine and neostigmine. 

Please proofread your paper, there are some minor mistakes. 

Undoubtedly, your work admirable considering the large amount of patients included, however it does not add new information to actual PONV development as you already underlined in conclusions. 

Author Response

Reviewer 1

Comment:

I am pleased to review your article and I admire your effort considering the enormous amount of data you processed. I would suggest to improve tables layout, because are really difficult to read and to interpret.

In my opinion, to better read and understand tables, you should sort your data by demographic characteristics, type of surgery and potential risk factors for PONV following Apfel score. In this way, a reader may understand quickly the differences between sugammadex, pyridostigmine and neostigmine. 

Please proofread your paper, there are some minor mistakes. 

Undoubtedly, your work admirable considering the large amount of patients included, however it does not add new information to actual PONV development as you already underlined in conclusions.

Answer:

Thank you very much for your comments.

We have sorted the tables according to your opinion.

We have corrected minor mistakes.

Reviewer 2 Report

The authors performed a propensity-matched retrospective cohort study on the effect of choice of the drug used for reversal of neuromuscular blockade on postoperative nausea and vomiting.  The manuscript reports a lower incidence of PONV with sugammadex when compared to neostigmine, but not when compared with pyridostigmine.

GENERAL COMMENTS

  1. The definition of PONV is a bit unclear. How was nausea or vomiting defined in the electronic medical record. Were specific fields searched? 

  1. Did you consider examining a group of patients who received no reversal drug or patients who did not receive a neuromuscular blocking agent?

  1. The use of the term “multicenter” is a bit misleading. It would be more specific to label it as a “5=hospital study.” Multicenter generally suggests more than one hospital or health system participating in a study.  The current study shares a single health system and data warehouse. 

  1. Additional information concerning the 5 hospitals should be presented. This is especially important since matching was not performed for hospital location. Conceivably, one hospital might use more sugammadex than the others, but might have a more conservative definition of PONV than the other hospitals.  Each hospital’s contribution of patients to the subject sample, preference for reversal drug,  and incidence of PONV is also not included. 

  1. Why were the neostigmine and pyridostigmine groups not directly examined? This is necessary because the effects of the two drugs are PONV, compared to sugammadex, were different. Odds ratios for PONV with neostigmine and pyridostigmine should also be presented. 

  1. A CONSORT-type flow diagram would be useful to outline inclusion/exclusions and the various groups.

  1. Sugammadex had a similar incidence of PONV as pyridostigmine. This is a bit surprising, given that the mechanism of action of pyridostigmine and neostigmine are similar. What might account for this finding?  This should be more fully examined in the discussion section. What confounding  factors might have been present?

  1. Figures 2 and 3 present very little graphical information and should be deleted. The odds ratio data can be incorporated into the text of the Results section.

SPECIFIC COMMENTS

p. 2, line 91-92: The inclusion/exclusion criteria 1) used no NMBRDs or 2) more than two types of NMBRDs should be specifically stated in the methods.

Table 1:  The data for ASA PS 1 for the after-matching sugammadex column appears to be incorrect.

p. 4, line 112: Italicize the section head to be consistent with the other sections.

p. 5, line 122-129: This section should be rewritten to include the data from the odds ratio graphs in the text. A comparison of PONV with neostigmine versus pyridostigmine is missing. Its inclusion would add to the internal consistency of the data presentation.

p. 6, paragraph 2 beginning on line 142: There are a half dozen or more studies comparing the incidence of PONV with sugammadex and neostigmine. I would suggest discussing these in a more organized fashion. These are best summarized by the Cochrane review by Hristovska (reference 20) although the Paech paper had not yet been published.  The paragraph could be better focused by 1) summarizing the meta-analysis finding of Hristovska, 2) commenting on the study designs and results, 3) commenting on how the current study adds to or is at odds with prior work.

p. 7, line 174: This sentence mentions that the use of pyridostigmine is limited, but pyridostigmine use was by far more common than the use of neostigmine and sugammadex in the current study. Why was there a predilection for pyridostigmine use in your subject sample?

p. 8, line 210: Insert the appropriate authors’ initials.

Figure 1A: in the column labels, NeoStigmine should be neostigmine.  It would be simpler to delete these and simply cite the odds ratio data in the text of the Results section.

Figure 1 needs a figure legend and definition of the y-axes.

Figure 1 does not have a figure legend, but is referred to as “Figure 1” in the text (p4, line 115).

Figure 2 is mislabeled as Figure 1 (p. 5, line 131)

Figure 3 is mislabeled as Figure 2 (p. 6, line 136).

Author Response

Reviewer 2

Comment 1.

The definition of PONV is a bit unclear. How was nausea or vomiting defined in the electronic medical record. Were specific fields searched? 

Answer 1:

Thanks for your comment. We defined PONV as nausea or vomiting within 24 hours after surgery and anesthesia. The medical record form of the five hospitals is the same, and there are fields to check nausea and vomiting. It can be found in the electronic medical record within 24 hours after surgery.

Comment 2.

Did you consider examining a group of patients who received no reversal drug or patients who did not receive a neuromuscular blocking agent?

Answer 2:

Thank you for your comment. In our study, few patients did not use a neuromuscular blocking agent during anesthesia. In most cases not using reversal drugs, only succinylcholine was used as a neuromuscular blocking agent, and surgery with only succinylcholine was almost a very short and specific operation. In addition, since neuromuscular monitoring was not applied in many patients, reversal drug was used in most patients with neuromuscular blocking agents. For these reasons, it was difficult to perform the analysis by matching with patients who did not use neuromuscular blocking drugs or cholinesterase inhibitors

Comment 3:

The use of the term “multicenter” is a bit misleading. It would be more specific to label it as a “5=hospital study.” Multicenter generally suggests more than one hospital or health system participating in a study.  The current study shares a single health system and data warehouse. 

Answer 3:

Thanks for your advice. We agree to your opinion. We changed the title of the paper form "Comparison of the effects of sugammadex, neostigmine, and pyridostigmine on postoperative nausea and vomiting: A multicenter matched cohort analysis" to "Comparison of the effects of sugammadex, neostigmine, and pyridostigmine on postoperative nausea and vomiting: A propensity matched study of five hospitals". We changed "multicenter" to "five hospital" in abstract and discussion as well.

Comment 4: 

Additional information concerning the 5 hospitals should be presented. This is especially important since matching was not performed for hospital location. Conceivably, one hospital might use more sugammadex than the others, but might have a more conservative definition of PONV than the other hospitals.  Each hospital’s contribution of patients to the subject sample, preference for reversal drug, and incidence of PONV is also not included. 

Answer 4:

Thank you very much for your opinion. That's very good advice. We haven't fully considered it. We believe that the quality of research will be better if hospital-specific data are added. Unfortunately, however, we do not include hospital-specific data in our basic data. Because the data we could obtain did not include hospital data, as well as data that could identify the patient, such as the patient's area of residence, hospital registration number and name, we did not consider the possibility of hospital-specific differences before the study began. A more complicated process is required to obtain data on the hospital where the patient was operated. It is impossible to apply for and obtain data from an institution that manages a clinical data warehouse in a limited time. Hospital-specific data cannot be included given the time limit of the revision. However, all hospitals included in the study are located in the metropolitan area. Also, since sugammadex is expensive, its relatively higher price than other reversal agents may affect the use of sugammadex. However, since South Korea has a national health insurance system, medical expenses are relatively low, so usage restrictions of sugammadex due to cost are small.

Comment 5:

Why were the neostigmine and pyridostigmine groups not directly examined? This is necessary because the effects of the two drugs are PONV, compared to sugammadex, were different. Odds ratios for PONV with neostigmine and pyridostigmine should also be presented. 

Answer 5:

Thanks for your comment. We fully agree with your opinion. We accepted your opinion and analyzed the PONV odds ratio of pyridostigmine to neostigmine after performing propensity score matching between the group using pyridostigmine and the group using neostigmine. 19,377 patients were included in each of the pyridostigmine and neostigmine matched patient groups. The unadjusted and adjusted odds ratios OR for PONV of the pyridostigmine group between the neostigmine and pyridostigmine groups were < 1.0 and were both statistically significant (unadjusted, OR: 0.23, 95% CI: 0.21–0.25, P<0.0001; adjusted for all variables, OR: 0.22, 95% CI: 0.20–0.24, P<0.0001; adjusted for all variables and propensity scores, OR: 0.22, 95% CI: 0.20–0.24, P<0.0001).

Comment 6:

A CONSORT-type flow diagram would be useful to outline inclusion/exclusions and the various groups.

Answer 6:

Thanks for your comment.  We also agree that flow charts will be useful for describing inclusion/exclusion outlines. So, I added a flow chart to Figure 1.

Figure 1. Flow chart, please see the attached.

 Comment 7:

Sugammadex had a similar incidence of PONV as pyridostigmine. This is a bit surprising, given that the mechanism of action of pyridostigmine and neostigmine are similar. What might account for this finding?  This should be more fully examined in the discussion section. What confounding factors might have been present?

 Answer 7:

Thanks for your good comments. In our study, neostigmine and pyridostigmine showed odds differences in the occurrence of PONV despite the similar structure and mechanism of action. We are considering several possibilities that could make such a difference. The possible reasons for these results are described in the discussion.

In this study, PONV occurrence of neostigmine was higher than the other two drugs. The difference of PONV occurrence between neostigmine and the other two drugs is difficult to consider as a difference in direct drug action mechanisms, because the neostigmine and pyridostigmine are similar in structure, and pyridostigmine shares neostigmine covalent binding to acetylcholinesterase and its lipid solubility[10,11]. However, some factors may cause the difference between neostigmine and the other two drugs for PONV. First, anticholinergic drugs administered concurrently with NMBRD may affect differences of PONV occurrence.[9] In this study, because sugammadex was not administered simultaneously with an anticholinergic agent, it was not investigated separately, but the preference of an anticholinergic agent administered at the same time may affect the difference in PONV. Atropine has been reported to have an antiemetic effect. [25,27] Second, because high doses of neostigmine may induce PONV [8,25,26], the administered dose of the drugs may affect the difference in the occurrence of PONV. Some clinicians frequently use full doses due to fast onset and short action time of neostigmine.[10] However, more research is needed on the factors that may cause differences in the occurrence of PONV between the two drugs.

Comment 8:

Figures 2 and 3 present very little graphical information and should be deleted. The odds ratio data can be incorporated into the text of the Results section.

Answer 8:

Thanks for your advice. The odds ratio data was incorporated into the text of the Results section.

SPECIFIC COMMENTS

Comment 1:

  1. 2, line 91-92: The inclusion/exclusion criteria 1) used no NMBRDs or 2) more than two types of NMBRDs should be specifically stated in the methods.

Answer 1:

Thanks for your comment. We have included the above two in the exclusion criteria of the method section.

  • Repeat surgery within 24 h after anesthesia
  • Lack of consciousness after surgery
  • Ventilator therapy after anesthesia
  • Nausea or vomiting before anesthesia
  • Insufficient data because the follow-up duration was shorter than 24 hours
  • Neuromuscular block reversal drugs (NMBRD; sugammadex, neostigmine and pyridostigmine) are not used.
  • Two or more types of NMBRD are used.

Comment 2:

Table 1:  The data for ASA PS 1 for the after-matching sugammadex column appears to be incorrect.

Answer 2:

Thanks for your comment. We have corrected what was wrong.

 0.01 -> 1156 (15.0)

Comment 3:

  1. 4, line 112: Italicize the section head to be consistent with the other sections.

Answer 3:

Thanks for your comment. We italicize the section head to match the other sections.

  1. Occurrence and odds ratio PONV

Comment 4:

  1. 5, line 122-129: This section should be rewritten to include the data from the odds ratio graphs in the text. A comparison of PONV with neostigmine versus pyridostigmine is missing. Its inclusion would add to the internal consistency of the data presentation.

Answer 4:

Thanks for your comment. We have included comparative data including the odds ratio of neostigmine and pyridostigmine in the text according to the comments.

The unadjusted and adjusted odds ratios OR for PONV of the pyridostigmine group between the neostigmine and pyridostigmine groups were < 1.0 and were both statistically significant (unadjusted, OR: 0.23, 95% CI: 0.21–0.25, P<0.0001; adjusted for all variables, OR: 0.22, 95% CI: 0.20–0.24, P<0.0001; adjusted for all variables and propensity scores, OR: 0.22, 95% CI: 0.20–0.24, P<0.0001).

Comment 5:

  1. 6, paragraph 2 beginning on line 142: There are a half dozen or more studies comparing the incidence of PONV with sugammadex and neostigmine. I would suggest discussing these in a more organized fashion. These are best summarized by the Cochrane review by Hristovska (reference 20) although the Paech paper had not yet been published.  The paragraph could be better focused by 1) summarizing the meta-analysis finding of Hristovska, 2) commenting on the study designs and results, 3) commenting on how the current study adds to or is at odds with prior work.

Answer 5:

Thanks for your comment. We have described the results of the meta-analysis, the design and results of the study, and any additions to our study.

[12]. In a systematic review, Hristovska et al. found that sugammadex was associated with a lower risk of PONV compared to neostigmine, and Gan Tong et al. suggested the use of sugammadex instead of neostigmine as NMBRD as a strategy to reduce the baseline risk of PONV. [20,21]. In a meta-analysis of Hristovska et al., including 10 randomized controlled trials, was performed to compare the efficacy and safety of sugammmadex and neostigmine regardless of dose, and 6 of 10 randomized controlled trials were included for comparative analysis of PONV. In their study, patients receiving sugammadex had 40% fewer adverse effects than those receiving neostigmine. In PONV, the relative risk of sugammedex to neostigmine was 0.52 (95% CI, 0.28-97). The results were similar to those of our study. However, due to the small sample size and limited evidence for the six cases included in the meta-analysis, the authors of the study suggested that a study with a large sample size and low risk of bias is needed in the future. Our study confirms and supports their findings by large sample size and reducing the bias with propensity score matching analysis.

Comment 6:

  1. 7, line 174: This sentence mentions that the use of pyridostigmine is limited, but pyridostigmine use was by far more common than the use of neostigmine and sugammadex in the current study. Why was there a predilection for pyridostigmine use in your subject sample?

Answer 6:

Compared to pyridostigmine, Neostigmine has a faster onset but shorter duration, five times higher potency and greater muscarinic action, but no difference in recovery from neuromuscular blockade. There is no specific reason, but in South Korea, the use of pyridostigmine is more common and preferred than neostigmine. (54.2 % vs 26.0%) The less frequent use of neostigmine compared to pyridostigmine seems to be due to past trends. (Anesthesia and Pain Medicine 2019;14(4):441-448.)

Comment 7:

  1. 8, line 210: Insert the appropriate authors’ initials.

Answer 7:

Thanks for pointing out our mistakes. We have inserted the appropriate author's initials.

Comment 8–12:

Figure 1A: in the column labels, NeoStigmine should be neostigmine.  It would be simpler to delete these and simply cite the odds ratio data in the text of the Results section.

Figure 1 needs a figure legend and definition of the y-axes.

Figure 1 does not have a figure legend, but is referred to as “Figure 1” in the text (p4, line 115).

Figure 2 is mislabeled as Figure 1 (p. 5, line 131)

Figure 3 is mislabeled as Figure 2 (p. 6, line 136).

Answer 8–12:

Thanks for your comment. We deleted Figure 1–3 and inserted the data into the text.

Reviewer 3 Report

The authors have presented a large patient analysis of the PONV effects of sugammadex, neostigmine and pyridostigmine.

Please reference in your discussion, the Gan et al, 4th PONV Consensus guidelines.

Please add in your discussion the limited information of pyridostigmine effects on PONV.

Please add more discussion why do you think there is no difference of PONV effect between sugammadex and pyridostostigmine. 

Author Response

Reviewer 3

Comment 1:

Please reference in your discussion, the Gan et al, 4th PONV Consensus guidelines.

Answer 1:

Thank you very much for your comments. We referenced to Gan et al, 4th PONV Consensus guidelines in discussion.

Line 185

Gan Tong et al. suggested the use of sugammadex instead of neostigmine as NMBRD as a strategy to reduce the baseline risk of PONV.

Comment 2:

Please add in your discussion the limited information of pyridostigmine effects on PONV.

Answer 2:

We have added information about pyridostigmine.

Line 185

Kang et al.[23] reported that sugammadex did not reduce the incidence of PONV compared to pyridostigmine in patients undergoing microvascular decompression, and the incidence of PONV decreased when administered simultaneously with panlonosetrone. It would be difficult to consider as a standalone effect of sugammadex.

Line 196–198

Pyridostigmine can cause PONV due to muscarinic side effects.[23]  A relative overdose of pyridostigmine compared to anticholinergic drugs can also trigger PONV.[19]

Comment 3:

Please add more discussion why do you think there is no difference of PONV effect between sugammadex and pyridostostigmine.

Answer 3:

We further compared pyridostigmine and neostigmine and showed that pyridostigmine had a lower incidence of PONV than neostigmine. So we focused on the reasons for the high PONV incidence of neostigmine and inferenced the possible reasons.

Line 199

In this study, PONV occurrence of neostigmine was higher than the other two drugs. The difference of PONV occurrence between neostigmine and the other two drugs is difficult to consider as a difference in direct drug action mechanisms, because the neostigmine and pyridostigmine are similar in structure, and pyridostigmine shares neostigmine covalent binding to acetylcholinesterase and its lipid solubility[10,11]. However, some factors may cause the difference between neostigmine and the other two drugs for PONV. First, anticholinergic drugs administered concurrently with NMBRD may affect differences of PONV occurrence.[9] In this study, because sugammadex was not administered simultaneously with an anticholinergic agent, it was not investigated separately, but the preference of an anticholinergic agent administered at the same time may affect the difference in PONV. Atropine has been reported to have an antiemetic effect. [25,27] Second, because high doses of neostigmine may induce PONV[8,25,26], the administered dose of the drugs may affect the difference in the occurrence of PONV. Some clinicians frequently use full doses due to fast onset and short action time of neostigmine.[10] However, more research is needed on the factors that may cause differences in the occurrence of PONV between the two drugs.
